# Features of Oil Spills Monitoring on the Water Surface by the Russian Federation in the Arctic Region

Artem Alekseevich Khalturin *, Konstantin Dmitrievich Parfenchik and Vadim Anatolievich Shpenst

Department of Electroenergetics and Electromechanics, Saint-Petersburg Mining University,
21st Line of Vasilievskiy Island, 2, 199106 St. Petersburg, Russia
* Correspondence: s195022@stud.spmi.ru

**Abstract:** Given that the recent rapid growth of offshore production, especially in the Arctic region of the Russian Federation, is causing increased concern about oil spills on the water surface, this issue is especially relevant and important today. These pollutants have a devastating impact on the world's marine biosphere. Therefore, effective and reliable methods and instruments must be used for operational spill detection in order to detect a remote oil spill. Several methods for oil spill monitoring and Russian developments in this area were described, including their features, advantages, and drawbacks. In cases when use in difficult Arctic conditions was anticipated, due to the harsh climate and ice-covered water surface, it was not always possible for spill detection instruments to be utilized. Despite this, such methods as radar, infrared, and ultraviolet were proven to be effective during this research. Ultimately, the combination of these methods returned the greatest volume of information to offshore platform staff about a detected oil spill. The information provided includes the spread area of the spill, the thickness of the leak, and the chemical composition of the oil.

**Keywords:** remote sensing; oil spills; the Arctic region; radar; infrared; ultraviolet; cosecance directional diagram

## 1. Introduction

An oil spill is a mass of oil that floats on the surface of a water body and is carried by the wind, currents, and tides. These can have detrimental effects on coastal ecosystems. There are numerous factors that can contribute to incidents of oil and petroleum product spills, including wear and tear on facilities, poor quality of repair and restoration, natural disasters, negligence in the workplace, acts of terrorism, and wars. An oil spill can happen at any point in the oil production and refining process, including during storage and transportation [1]. Potential sources of oil spills include well blowouts during subsea exploration or production, releases or spills from subsea pipelines, leaks from onshore storage tanks or pipeline leaks, and accidents involving vessels [2,3].

The impact of oil spills in the Arctic is particularly serious due to the adverse effects they could have on the delicate ecosystem of the region and because it would be especially difficult to detect and track oil seepage beneath the ice [4]. Conditions such as moving ice, low temperatures, limited visibility or complete darkness, high-speed winds, and extreme storms increase the risk of accidents or errors that can lead to oil spills. The hazard of environmental and economic damage from major oil spills in Arctic waters can be significantly reduced by the development of faster and more reliable sensing techniques [5].

Rapid and effective detection of oil pollution on the water surface is possible with contemporary remote monitoring methods. Broadcast data about the volume, area, and even chemical composition can then be obtained.

## 2. Oil Pollution of the World's Waters Today

The oil market grew rapidly in the mid-20th century. Large tanker ships capable of carrying over 100 thousand tons of raw materials began to appear, along with underwater

pipelines and floating oil production platforms. However, as technology continued to develop, the human factor also gained in importance. One wrong action can lead to a genuine environmental catastrophe.

## 2.1. Oil spills Caused by Marine Transportation

In the 1980s, several explosions occurred in the Caribbean Sea due to a collision between two tankers, the Atlantic Empress and the Aegean Captain. This led to over 2.1 million barrels of oil falling into the sea. It is believed that the coastline was affected because the catastrophe occurred in the open sea. Marine flora and fauna suffered significantly [6].

However, the number of oil spills from tankers has gone down a lot in the last few decades. Since 1970, spills exceeding 7 tonnes have decreased by over 90% [7].

Table 1 presents the major shipwreck spills that have occurred since records began to be kept.

**Table 1.** Major ship spills [8].

| № | Shipname | Date | Location | Spill Size (Million Barrels) |
|---|---|---|---|---|
| 1 | Atlantic Empress | 1979 | Off Tobago, West Indies | 2.1 |
| 2 | ABT Summer | 1991 | 700 nautical miles off Angola | 1.9 |
| 3 | Castillo de Bellver | 1983 | Off Saldanha Bay, South Africa | 1.8 |
| 4 | Amoco Cadiz | 1978 | Off Brittany, France | 1.6 |
| 5 | Haven | 1991 | Genoa, Italy | 1 |
| 6 | Odyssey | 1988 | 700 nautical miles off Nova Scotia, Canada | 1 |
| 7 | Torrey Canyon | 1967 | Scilly Isles, UK | 0.9 |
| 8 | Sea Star | 1972 | Gulf of Oman | 0.8 |
| 9 | Sanchi | 2018 | Off Shanghai, China | 0.8 |
| 10 | Irenes Serenade | 1980 | Navarino Bay, Greece | 0.7 |
| 11 | Urquiola | 1976 | La Coruna, Spain | 0.7 |
| 12 | Hawaiian Patriot | 1977 | 300 nautical miles off Honolulu | 0.7 |
| 13 | Independenta | 1979 | Bosphorus, Turkey | 0.7 |

## 2.2. The Arctic Incidents

For the purposes of this article, the Arctic border is defined as the region where the average temperature in the warmest month (July) is below 10 °C. Despite efforts being made to prevent spills in the area, further statistics indicate that the experience of previous major oil and petroleum product spills is unfavorable.

In the Northwest and North Slope regions, small crude oil and refined oil spills less than 50 barrels in volume are fairly common [9].

The largest spill in Alaska occurred on 24 March 1989, when an Exxon Valdez tanker owned by Exxon, one of the largest oil companies, collided with a reef off the Alaskan coast and ran aground in Prince William Sound. The estimated amount of oil spilled was around 260,000 barrels, and the damage to the region's environment and local fisheries was by all accounts the worst and most lasting in the history of such disasters [10].

The largest spill in the U.S. Arctic was caused by the bulk carrier Selendang Ayu, which grounded near Unalaska in the eastern Aleutian Islands in December 2004. The ship leaked over 8000 barrels of fuel as well as its cargo of soybeans [11].

From 1981 to 1983, a series of small, experimental oil spills were conducted on the northern tip of Baffin Island in Canada. An estimated 179,000 barrels of Bunker C was released when the Arrowran aground in Chedabucto Bay, Nova Scotia, in 1970. Major changes to the legal framework related to oils spills in Canada were caused by this spill [12].

Between 1999 and 2009, no major oil spills occurred in the Russian Federation, but numerous smaller spills did. In May of 2020, a large amount of diesel fuel was released from a holding tank near Norlisk, which supplied a power plant [13].

Major oil spills have also occurred in the Arctic waters of European nations. In 1977, the Bravo platform experienced an oil and natural gas blowout, resulting in the first

major oil release in the North Sea. Before the well was capped, it was estimated that over 200,000 barrels of oil had been spilled [14].

On 5 January 1993, the tanker Braer made an emergency landing in the south of the Shetland Islands due to a storm and engine failure. It was heading from Norway to Canada with a cargo of 621,000 barrels of Gullfaks crude oil, which leaked into the sea. The Braer oil spill is one of the largest in terms of the amount of oil spread into the environment [15].

The tanker Antonio Gramsci disaster occurred in 1987, when the tanker ran aground off the south coast of Finland, spilling between 89,000 barrels and 107,000 barrels of crude oil [16,17]. The Soviet tanker Volgoneft 263 collided with a West German cargo ship called Betty in 1990. This caused 6300 barrels of waste oil to be released off the Swedish coast [18].

### 2.3. Energy Pipelines Accidents

On 11 May 2021, the oil pipeline of the Osh field was depressurized 300 m from the bank of the Kolva River, Northern Russia. The majority of the mixture of oil, water, and sand that was spread on the soil occupied the natural lowland near the place of the leak. However, approximately 10% of the spill ended up in the Kolva—an oil slick on the water was used for the leak detection. On land, the spill was swiftly contained [19].

### 2.4. Oil Spills Caused by Petroleum Exploration and Production

Over the years, there have been several major oil spill accidents caused by offshore operations during oil production in the sea.

On 20 April 2010, there was an explosion on the Deepwater Horizon oil platform in the Gulf of Mexico in the Macondo field. This resulted in the catastrophic oil spill. The Deepwater Horizon platform was a modern engineering facility that used the latest technology to organize and conduct offshore drilling. There were 100 million barrels of potential reserve. The Deepwater Horizon disaster was the culmination of a chain of events. After thirty-six hours from the beginning of this catastrophic event, the Deepwater Horizon platform sank, ending up on the seabed at a depth of 1500 m. It became apparent three days after the events of April 20th that the extent of the accident was 250 square kilometers, as the area of the oil slick reached that size. After growing for a week, it moved towards the U.S. coast and reached 80,000 square kilometers [20,21].

Another noteworthy accident occurred on 12 December 2007, when approximately 4000 m$^3$ of oil spilled into the North Sea from an offshore oil platform located about 200 km west of the city of Bergen in Norway. The Statfjord field is one of Norway's largest oil fields and is located near the border between British and Norwegian territorial waters. The spill occurred when the shuttle tanker Navion Britannica was pouring oil [22].

Table 2 provides a list of the major oil spill accidents that occurred since records began to be kept.

**Table 2.** Oil spills greater than one million barrels since 1980 caused by offshore exploration and production [3].

| № | Name | Date | Location | Cause | Spill Size (Million Barrels) |
|---|------|------|----------|-------|------------------------------|
| 1 | Gulf War | January 1991 | Persian Gulf, Middle East | Iraq–Kuwait War | 5–8 |
| 2 | Deepwater Horizon | April–July 2010 | Gulf of Mexico, US | Wellhead blowout | 4–5 |
| 3 | Ixtoc I | June 1979–March 1980 | Bay of Campeche, Gulf of Mexico | Exploratory well blowout | 3.3–3.5 |
| 4 | Fergana Valley | March 1992 | Uzbekistan | Oil well blowout | >2 |
| 5 | Nowruz oil field | February–September 1983 | Persian Gulf/Iran | Iraq–Iran war | >1.9 |
| 6 | Production Well, D-103 | August 1980 | Tripoli, Libya | Well blowout | ~1 |

## 3. Features of Oil Spills Monitoring in the Arctic Region

### 3.1. The Potential of the Arctic Territories

The importance of the Arctic territory's development is increasing. This is a promising tourist destination for a variety of countries. Particularly protected natural areas are home to unique creatures and indigenous small-numbered peoples.

Different countries' priorities in the Arctic region include the efficient use of raw materials and the logistics potential [23–27]. There is a significant resource base of minerals and hydrocarbons. The energy transition depends on their extraction being a driver of economic development. There are numerous major mineral producers and oil and gas companies operating in the region, for example LLC Varandeyskiy Terminal, LLC Gazprom Neft Shelf, public joint-stock metal and mining company Norilsk Nickel, Arctic Mining Company Ltd. (Norilsk, Russia)., China National Offshore Oil Corporation Ltd. (Beijing, China), and China National Petroleum Corporation, ConocoPhillops Alaska Inc. (Beijing, China), Exxon Mobil Corporation (Irving, TX, USA), DEA Norge (Stavanger, Norway), Equinor (Stavanger, Norway), Agnico Eagle Mines Ltd. (Toronto, ON, Canada), et al. [28–32].

Industrial production has great potential for growth, which will lead to drastic changes in the region and complicate logistics schemes [33]. The effects of climate change in the Arctic pose serious threats and challenges, but also open up additional opportunities for economic activities [34–36]. The Northern Sea Route, the main transport artery between the Atlantic and Pacific Oceans, has the potential to become a high-capacity transit shipping route [28].

With an increasing number of ships navigating through ice-filled waters, there is a greater risk of oil spills in these areas. The spilling of oil by various means during its storage and transportation has become inevitable, since a large amount of oil is being used daily [37,38]. On average, over 80% of Arctic oil resources are concentrated offshore, which makes developing a spill-response system challenging given the difficult ice and weather conditions [39–41].

Potential sources of subsurface oil include a pipeline leak, the leakage from a submerged tank or vessel, or a natural seep (Figure 1). To preserve the region's unique and vulnerable environment, it is necessary to develop an efficient remote monitoring system.

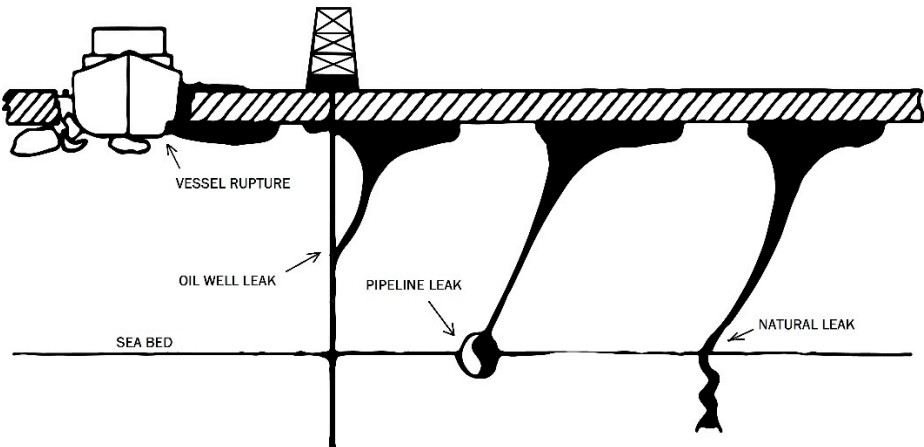

**Figure 1.** Potential sources of subsurface oil: vessel rupture; oil well leak; pipeline leak; natural leak (created by authors).

### 3.2. Oil on Ice

Sea ice is interacting with oil in many forms. Liquid petroleum can be found in a network of pores and drainage channels. Oil that becomes trapped on or beneath the surface of the ice usually does not spread over long distances and is localized at the site of the accidental spill. Depending on the weather conditions, the ice layer may be in a rapid growth phase, stationary, or in the process of breaking up, which would allow the oil on the surface to be surrounded by ice, at the ice–water interface, or trapped in the growing ice

layer [41]. Fast ice, which has frozen to a sea floor or a coast, will not move with currents or winds. It is therefore unlikely that oil that spills on or under fast ice will drift. Pack ice floats freely on the surface, and oil spilled here will drift with it.

Furthermore, oil can get trapped under the ice. First-year growing sea ice can completely encapsulate oil. When oil is released into the sea, it is subject to the usual weathering processes, although it is often influenced by the low temperature and the presence of ice. Between 2 and 10 L of oil are estimated to enter the sea ice pore space per square meter of ice after a spill [42]. Cold temperatures hinder evaporation, and the lack of waves restricts dispersion. The absence of waves also impedes emulsification. This can increase the pollutant volume by up to five times [43].

Strong winds, poor visibility, unpredictable weather conditions, freezing, and drifting ice can be major obstacles to a prompt response for spills and can be a factor that reduces the effectiveness of oil spill response measures. In order to identify pollutants in the oil and ice mixture, it is necessary to have a big toolbox. When the oil is trapped beneath the ice sheet, new oil-under-ice detection systems are required. It is important to be prepared for a swift and coordinated response, using the most effective strategies, for minimizing the impact of the spill.

## 4. Existing Methods and Devices for Oil Spill Monitoring

There are many methods available today for monitoring oil spills on land and on the water surface. Below, the most used methods and instruments are described. A comparative analysis is provided for finding the most promising methods of spill detection.

The criteria for evaluating the best methods were:

- Speed (of on-time detection of the spill and prompt alarm of the incident);
- Accuracy (high-quality, close-to-reality monitoring data);
- Universality (the ability to use this method everywhere, under any conditions: on land, on water, etc.);
- Cost-effectiveness (cheapness of the method, not to the detriment of its quality);
- Individual available positive aspects of the method.

There are two types of monitoring methods that are globally used: laboratory (physico-chemical) and in situ.

Among the main physico-chemical methods are noted:

- Gravimetric analysis;
- Infrared spectrometry;
- Fluorescence spectroscopy analysis;
- Gas chromatographic analysis.

Among the main means of remote monitoring there are noted:

- Aircrafts;
- Unmanned aerial vehicles (UAVs);
- Satellites;
- Radars.

The essence of these methods is now being examined.

### 4.1. Gravimetric Method

This method is usually used for the analysis of waters with an extremely high concentration of hydrocarbons. The main idea of the gravimetric method is to extract oil components from a sample of polluted water with low-polar solvents, followed by purification of the extract from polar organic substances using special sorbents. During the evaporation of the extractant, the residue, or free carbon, is typically weighed.

Figure 2 shows the composition of the installation for determining the proportion of free carbon in accordance with Russian Standard GOST 26564.2-85 Refractory Silicon Carbide Materials and Products. The methods for determining free carbon come from Reference [44].

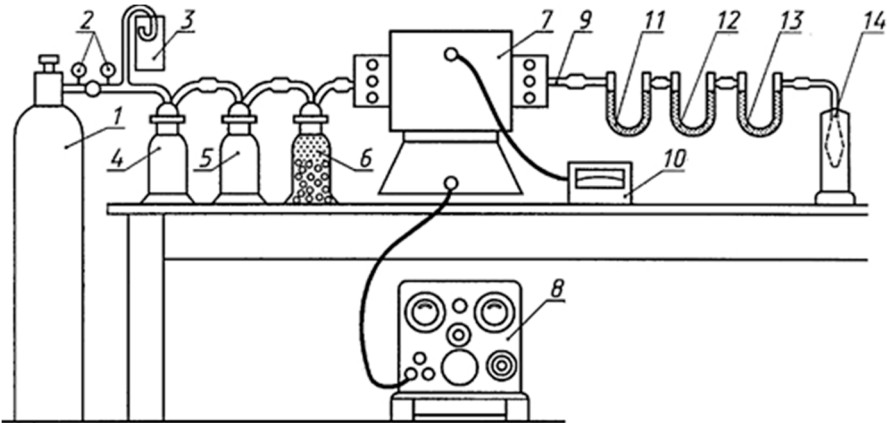

**Figure 2.** Equipment used to determine the proportion of free carbon [44].

Installation composition: 1—oxygen cylinder; 2—pressure-reducing valves; 3—pressure gauge: 4—a flask with glass wool; 5—a flask with a ring of iron and sulfuric acid; 6—a flask filled with 223 sodium lime and silica gels and glass wool gaskets; 7—a tubular furnace with electric heating; 8—an adjustment transformer: 9—gas-tight ceramic tube with a length of 800 mm and an internal diameter of 18 mm; 10—measuring device temperature; 11—C-shaped tube filled with silica gel; 12—shaped tube, ia 2/3 filled with sodium lime, with glass wool gaskets; 13—C-shaped tube, 2/3 of the iatronic lime and ia UZ silica gel, with glass wool gaskets; 14—bubble counter.

Since this method does not require preliminary calibration or verification of measuring instruments, or taking standard samples, it can be said to be one of the most accurate methods in the field of analytical chemistry.

The main disadvantage of the method is that it cannot be used at extremely low concentrations of hydrocarbons in water, since the measurement range starts at 0.35 mg/dm$^3$.

Nevertheless, the disadvantages can be overcome with a more modern method of two-dimensional gas chromatography (GC×GC) with mass spectrometry detection. The method is still being studied and improved. Over the past decade, the approach has evolved with applications in many fields and more clearly defined research questions. Recently, several novel flow modulator designs have been introduced in terms of instrumental advances. Despite the fact that this form of modulation still accounts for a small portion of the total GC×GC practice, an increase in its deployment is expected in the future.

Data analysis and handling and extrapolation of useful information from the information-rich 2D chromatograms, are still some of the biggest challenges, especially in untargeted analysis.

The detection of mass spectrometry is an important ally for GC×GC in continuing its journey through the maturation, expansion, and establishment of such a high-resolution multidimensional technique [45].

### 4.2. Infrared Spectrometry Method

According to Standard GOST R 51797-2001, the method of infrared spectrometry (hereinafter IR spectrometry) is approved for use in determining the concentration of petroleum products in drinking water [46].

When using the IR spectrometry method, the main stages of analysis include:

- Extraction of oil from the sample taken with an organic solvent;
- Purification of the extract from polar compounds using the column chromatography method;
- Registration of the intensity of the absorption spectrum of C-H bonds in the range of wave numbers 2700–3150 cm$^{-1}$;
- Determination of the concentration level of oil and petroleum products by the area of the spectrum or optical density.

It is worth noting that this method has several advantages, including its low dependence on the specific type of oil that caused the contamination of drinking water. It is therefore possible to jointly determine the content of volatile and non-volatile components.

The IR spectrometry method has a significant disadvantage due to the use of highly toxic, toxic substances as extractants (e.g., hladon 113, carbon tetrachloride, etc.).

### 4.3. The Fluorescence Spectroscopy Method

The process involves the extraction of petroleum products with hexane. If necessary, this extract is purified, and then the intensity of its fluorescence is measured. The cause of the observed fluorescence intensity is optical excitation.

The main advantages of this method are speed, high accuracy, and sensitivity, as well as a small sample volume requirement.

However, the fluorescence spectroscopy method is not suitable for widespread environmental control, since only aromatic carbon bonds are formed in the analytical signal.

The proportion of these hydrocarbons depends on the nature of petroleum products and can be very small. Therefore, there is a possibility of receiving false results [47].

### 4.4. Gas Chromatographic Method

One of the most promising methods of analysis is based on the oil products that were found in the sample. The extract obtained from polar compounds and purified is analyzed using a gas chromatograph.

The total area of the chromatographic peaks of hydrocarbons serves as a quantitative analytical signal. This method was developed and successfully applied in the laboratories of Mosvodokanal for the control and quality of drinking water between 1985 and 1995. Since then, it has been certified by the State Standard of the Russian Federation and has become one of the most reliable and informative ways to detect petroleum products in all kinds of waters (e.g., sewage, natural, drinking water, etc.).

The gas chromatographic method is not only capable of determining the total content of petroleum products in water, but also of identifying individual hydrocarbons. This can then be used to identify the source of pollution.

The above-mentioned basic physicochemical methods of monitoring oil spills provide a measure of the scale of the disaster (the level of petroleum products in water from a specific sample taken in a specific reservoir). Thanks to them, it may be possible to realize the full scale of a man-made accident, for example, the spill area, contaminated territories, damage caused, etc.

### 4.5. Aircrafts

The cheapest method of monitoring oil industry facilities is by aircraft. One of the greatest advantages of using the aircraft for monitoring purposes is the ability to use it continuously, even in harsh climatic conditions.

The operation of the aircraft includes:

- Continuous monitoring of oil production systems in the "online" mode;
- Use of a thermal imaging system built into the aircraft to locate possible oil leaks;
- Development of a digital map of onshore oil production facilities;
- Obtaining and processing photographic plans of territories;
- Prompt detection of possible illegal pipeline tapping;
- Monitoring of bush sites [48].

One of the most innovative and effective methods for using aircraft today is monitoring using unmanned aerial vehicles (also known as UAVs). The received frames from the UAV are shown in Figures 3 and 4.

Using UAVs makes it possible to detect unauthorized work with equipment and oil production facilities (e.g., tie-ins into an oil pipeline, landfills formed, work in restricted or protected areas, etc.).

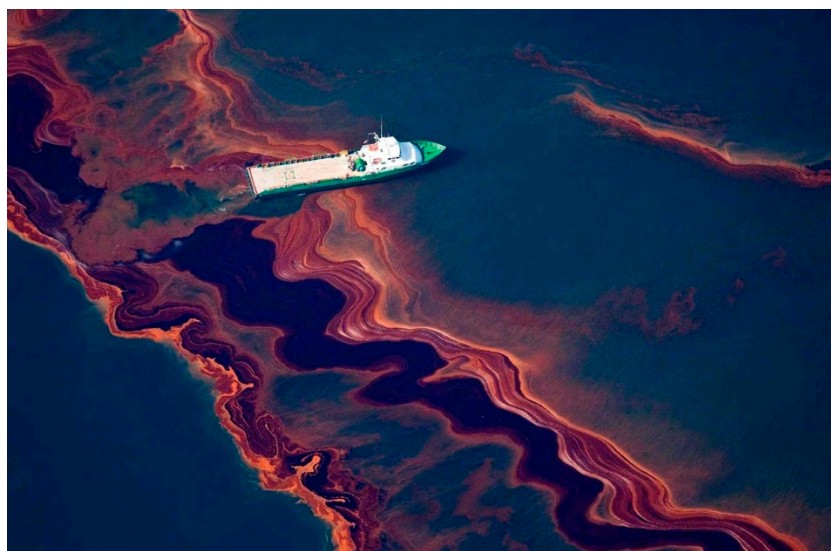

**Figure 3.** Example of an oil spill image obtained from a UAV [49].

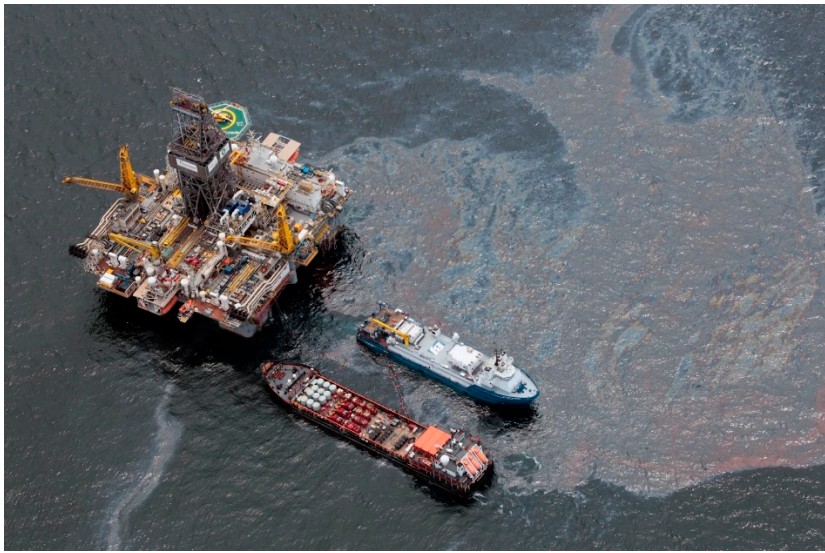

**Figure 4.** Example of an oil spill image obtained from a UAV [50].

UAVs, as well as other unmanned vehicles, can be of an airplane or helicopter type. These types are divided into the following classes (Table 3).

**Table 3.** Classification of UAVs [51].

| Type of UAV | UAV Weight, kg | Radius of the Action Zone, km |
|---|---|---|
| Micro and mini short range | To 7 | From 25 to 40 |
| Small radius, light | From 7 to 50 | From 15 to 115 |
| Medium radius, light | From 50 to 100 | From 110 to 300 |
| Medium | From 100 to 300 | From 170 to 1000 |
| Medium—heavy | From 300 to 500 | From 150 to 1000 |
| Medium radius, heavy | From 500 to 1000 | From 75 to 300 |
| Large radius, heavy | Above 1000 | Above 300 |

Today, the use of UAVs is commonplace both in private and large oil and gas companies. For example, this technology has made it possible to improve the efficiency of operation and maintenance of pressure pipes.

Over 15 thousand kilometers of oil pipelines were examined with the help of Izhevsk-made UAVs "ZALA 421–16EM" and "ZALA 421-16E". Up to 200 km of pipelines are monitored during one drone flight.

For the past few years, the Rubezh-30 type UAV has been successfully operated in the territory of Venezuela (Guarico State, San Juan Los Mortos airfield) in order to monitor oil industry facilities. The prototype was developed by the Kazan-based company 'Aarokon'.

Another example of the successful use of UAVs was the implementation of an aerial monitoring system at JSC Samotlorneftegaz facilities in April 2012.

In this case, the UAV was used to detect leaks and illegal taps into the pipelines and to monitor oil spills on land and water [48].

The main benefits of using a UAV are:

- Affordable cost;
- High-quality terrain survey followed by the creation of a digital terrain map using modern software;
- Through aerial surveillance, loss and leakage of fossils can be quickly and effectively controlled, saving time and increasing the effectiveness of the monitoring itself ring itself;
- There is minimal damage to the environment due to low fuel consumption and the lack of an infrastructure;
- The cost of using different aircraft for monitoring is presented in Table 4. The economic effect of using UAVs is shown in the graph in Figure 5.

**Table 4.** Classification of UAVs [51].

| Type of Aircraft | Costs per Hour of Flight (Relevant for November 2022) |
| --- | --- |
| Drone | ≈100 USD |
| Helicopter Robinson R-66 | from 930 USD |
| Mi-8T helicopter | from 1900 USD |

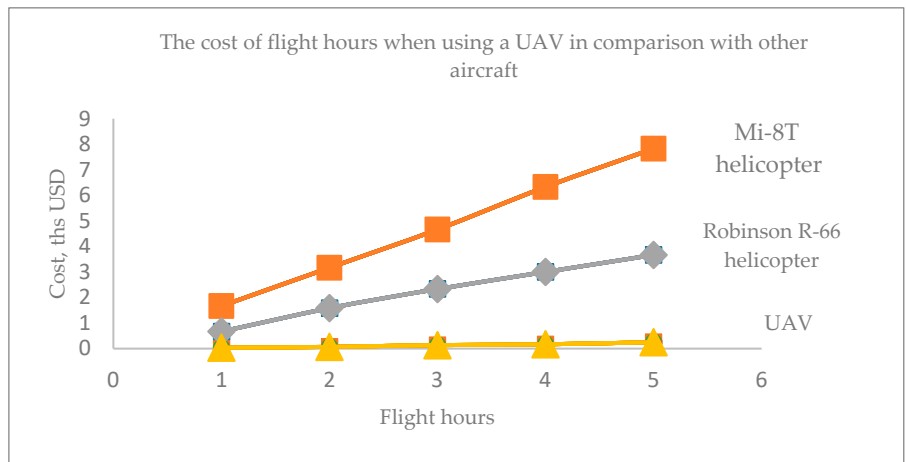

**Figure 5.** The cost of flight hours when using a UAV in comparison with other aircrafts [51].

This effect, given above, is explained quite simply by the fact that manned aviation requires expensive infrastructure, such as landing pads, refueling stations, control rooms, special, trained personnel, etc.

A UAV also has advantages in security purposes over other aircraft, since the failure of the drone will not entail huge financial and human losses. A plane crash can lead to fatalities.

*4.6. Radar Method*

The radar method is also currently considered to be an effective method of remote monitoring for surface water..

The main advantage of using airborne or stationary radar stations is their all-weather capability, as well as the ability to obtain information any time of day. This is due to their

wide field of view, which can range from tens of kilometers when probing using aviation to hundreds of kilometers when using space carriers.

The dependence between the characteristics of radio signals reflected from the sea and the parameters of surface waves is what determines the method of monitoring the sea surface (Figure 6) [52,53].

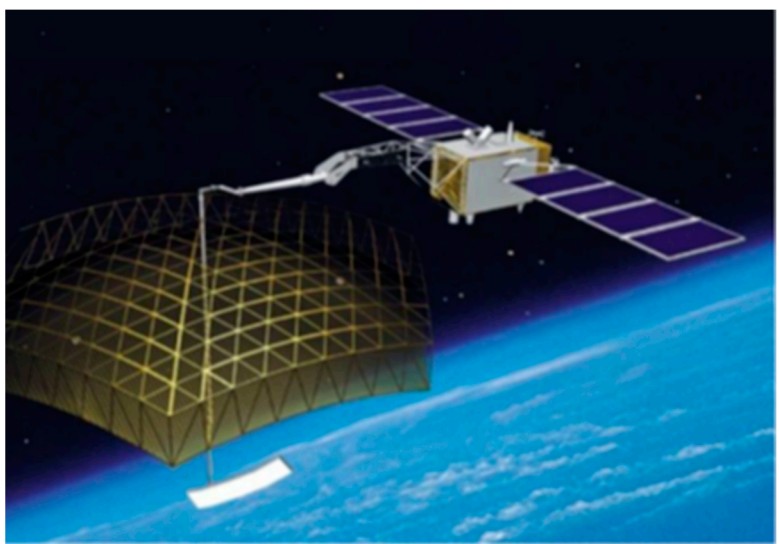

**Figure 6.** Image of the small spacecraft "Condor-E" with the installed radar [54].

The implementation of this method may take several forms:

- A radar on an artificial Earth satellite.

The essence of this method is to irradiate the studied area of the water surface with radio pulses, then to receive and register the echoes reflected during irradiation. The received amplitudes of the reflected echoes are then converted into specific effective scattering area (SESA) values for each element of the radar's spatial resolution. The SESA of the specific element of the radar's spatial resolution is reflected in the radar image (Figure 6).

The finishing stages of the radar's formation include filtration, segmentation, and processing to isolate dim areas of the radar that indicate the presence of oil pollution (spots) on this part of the sea surface [55].

The main disadvantages of this method of radar monitoring with satellite are:

- The radar may be susceptible to weather conditions that could cause distortion of the received radar images of the surveyed area;
- Since the satellite on which the radar is installed is in constant motion in orbit, the frequency of monitoring depends on how far the satellite is from the Earth. To achieve effective operational monitoring, it will be necessary to involve several satellites and coordinate their parallel work.

- Vessel radar.

When the vessel radar is activated, radio pulses with horizontal polarization are emitted into the sea surface, and the reflected echo signals are received, registered, and processed.

If the SESA value exceeds the threshold value, it signals the presence of oil pollution on this section of the sea surface. The vessel's radar appearance is shown in Figure 7.

One of the limitations of this method is that only a small radius of contamination detection can be distinguished, since everything here directly depends on the height of the radar installation. Usually, the radius of the studied surface of the vessel radar does not exceed 1 km. Therefore, for complete monitoring in the case of a large oil spill, several vessels will need to be involved at once [57].

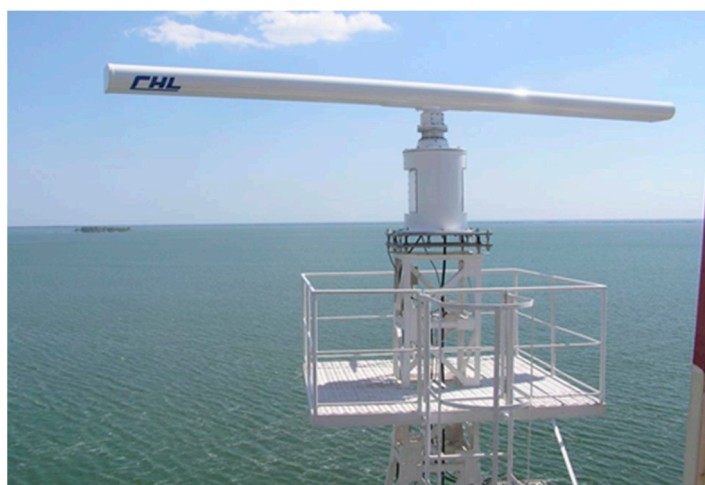

**Figure 7.** The appearance of the vessel radar [56].

- Coastal radar.

The main advantages of using a coastal radar are its efficiency and continuous ability to receive and process information. Small observation areas do not require a large radar range. This type of radar will have sufficient energy potential to detect oil pollution on the sea surface (Figure 8).

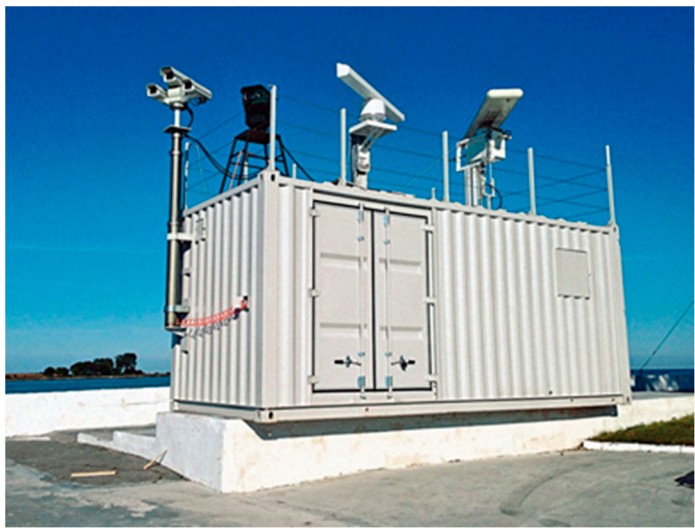

**Figure 8.** Appearance of the coastal radar [58].

Due to the low reflectivity of the water surface, the quality of the radar information requires a high-energy potential. This will correspond to the required probabilities of correct detection and false alarm [59].

*4.7. Conclusion on the Analysis of Oil Spill Monitoring Methods*

Considering the main criteria for the best method discussed at the beginning of this section, the radar method was chosen as a method to monitor the sea surface using ship, shore or stationary radar.

Additionally, to improve the quality monitoring of oil and gas industry facilities, a symbiosis of several types of capture can be used at once:

1. To monitor pipeline routes, video capture using a television camera is used. Operators can quickly monitor the current situation by watching video broadcasting in the "online" mode;

2. Using a digital camera allows for photos of the highest resolution to be obtained, which can then be subjected to spectrometry processing to identify problem areas (e.g., corrosion, excessive heating, etc.);
3. The location of an oil spill may be detected by filming in the IR spectrum, since the areas with a different shade in the frames will have more thermal radiation of oil.

## 5. The Effectiveness of Remote Monitoring Methods for the Arctic Region

If the volume of oil released into the sea is unknown, other methods can be used to estimate the amount from the surface area and thickness. The SAR satellite system, for example, could be used to monitor hundreds of small oil spills in the Mediterranean Sea each year [60].

A geographic information system (GIS) is a computer monitoring system capable of storing, analyzing, and sharing geographical information. It is possible to detect oil spills with this data [61]. Aircraft and satellite imagery can usually be used to detect and monitor oil spills. Oil spill detection by synthetic-aperture radar (SAR) is based on the dampening effect oil has on the surface waves of seawater. A SAR image classification can reveal three different classes for an oil spill: the spill area in the center surrounded by a high-pollution area, and the outer layers of a low-pollution area. When winds are very light, no SAR signal is received from the sea, and thus no oil slicks can be seen [62]. The unpolluted water surface shows higher surface roughness than the oil slick, which results in an increased back scattered signal [63].

There are several remote-sensing devices available for oil spill detection, including sensors consisting of frequency-modulated continuous wave radar, ground-penetrating radar (GPR), infrared photography, acoustics methods, thermal infrared imaging, airborne laser fluorosensors, airborne and spaceborne optical sensors, and airborne and spaceborne synthetic-aperture radar. Remote sensing devices may help to identify minor spills before they cause widespread damage. Remote sensors should be fully functional both during the day and at night. Researchers suggest that a combination of airplanes and satellites equipped with SAR sensors is the most effective and valuable tool for identifying oil spills. The steps involved in detecting an oil spill using SAR include radar image pre-processing, post-processing to identify the type and thickness of the spill, and finally, identification of the spills' location.

In November 2002, an oil spill of 11,000 tons occurred off the coast of Galicia in Spain, as reported by P. Montero [63]. Fly-overs were used to monitor the spill's movement, and its path was predicted by different models. Several helicopters and ships were used to monitor the spill, with the assistance of some volunteers and fishermen. Data obtained from fly-overs were entered into a GIS to obtain a picture of the situation. Assuming that the surface wind speed is 3%, the track of the oil spill can be obtained [63].

The aircrafts used for monitoring an oil spill must have good downward visibility, good radios for direct communications with vessels or ground personnel, and a global positioning system (GPS). The length and size of the spill can be determined by knowing the speed of the aircraft and the time taken to fly over it. The thickness and volume per square kilometer can be determined with the help of microwave radiometer.

The oil film thickness can be determined using a microwave radiometer because of the higher brightness temperature in the oil-contaminated area. The thickness of the oil film affects the temperature difference in the microwave range. Thus, oil here acts as a matching layer between the surface of water with a high dielectric constant and the surface of air with a low one. Increasing the thickness of the oil layer increases the brightness temperature. After this, it passes through alternating minima and maxima [64].

Previous studies have demonstrated that no single sensor is capable of detecting oil under all ice conditions. There are both advantages and limitations to each sensor [65]. Certain sensors may complement each other in terms of their ability to resolve oil thickness or swath width. Future operational systems are likely to employ suites of different sensors

operating from various platforms under, on, and above the ice surface to provide the means to detect oil in a range of ice environments at different times of the year.

The ice-coupled GPR is able to detect both fresh and encapsulated oil throughout the ice growth. Additionally, the airborne GPR is able to detect encapsulated oil when the ice is cold.

The optical instruments of monitoring (cameras) are able to determine the oil spill near the top of the bare ice, but they are unable to determine the oil underneath a thin layer of snow. For spotting oil among other dark targets, there is the potential of using near-infrared spectra. A thermal IR sensor can be used to detect the heating of oil at the surface of the ice.

There was a one-centimeter layer of oil that blocked the passage of light through the ice (Figure 9). The ice layer almost does not let in light, which makes it possible to determine oil using cameras and radiometers, even in the case of ice build-up under the oil layer. It has become possible to detect oil under a layer of ice of five to six centimeters with the usage of a fluorescence polarization sensor (Figure 10).

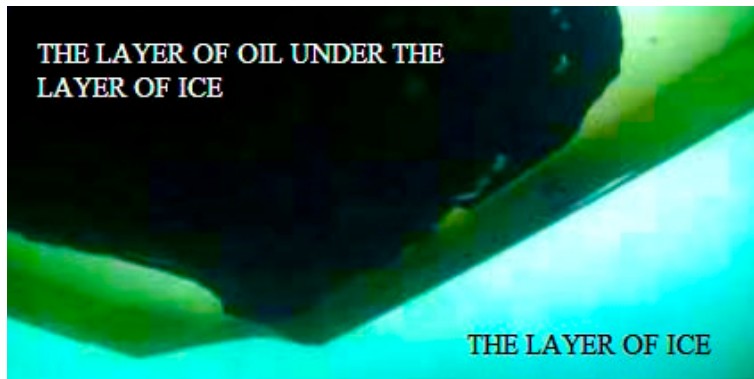

**Figure 9.** Image of oil below the layer of ice (looking from below) [66].

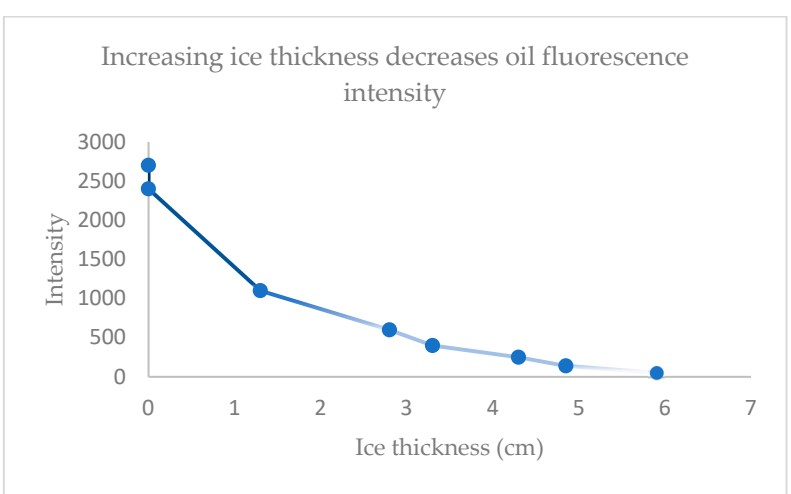

**Figure 10.** Graph of the effect of ice thickness on the intensity of oil fluorescence [66].

There are similar results of multipole multipath radars. With them, it is possible not only to determine the oil pollution itself under the ice layer, but also to get a rough estimate of the oil layer's thickness. (Figure 11). Radars operating in the frequency range from 200 to 400 kHz are capable of detecting oil inclusions beneath a layer of ice of about 6 cm [66].

In the Table 5 the 'P' rating indicates that there are conditions under which the system is expected to work and others under which it is expected to fail. P2 rating indicates that performance cannot yet be fully assessed. N/A means that sensors are not relevant in this situation, for example, using below-ice sensors to find oil on the surface [66].

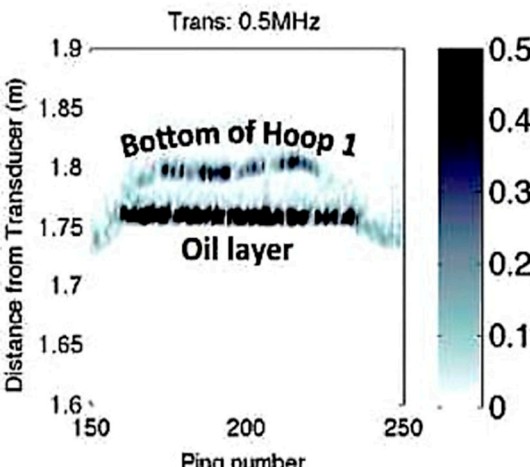

**Figure 11.** A transect under-hoop after an oil injection. The return from the base of the oil is the dark flat return, and the return from the ice is the lighter color above and to the sides of the oil [66].

**Table 5.** Predicted operational characteristics of the devices used based on the results of laboratory studies and computer modeling [66].

| Location | Airborne | | | | | On Ice | Below Ice | | |
|---|---|---|---|---|---|---|---|---|---|
| Sensor | GPR | FMCW | Optical | FP | IR | GPR | Optical | FP | Acoustic |
| **Fall-Winter-Spring** | | | | | | | | | |
| Exposed oil on ice | Y | P2 | Y | Y | P2 | Y | P2 | N/A | N/A |
| Snow covered oil on ice | Y | P2 | N | N | N | Y | P2 | N | N |
| Fresh oil under ice or with up to 6 cm (encapsulation) | P2 | P2 | N | N | N | Y | Y | Y | Y |
| Encapsulated oil (more than 6 cm new growth) | P | P2 | N | N | N | Y | P | N | P2 |
| **Summer** | | | | | | | | | |
| Exposed oil on ice | P2 | P2 | Y | Y | Y | N/A | N/A | N/A | N/A |

For all of the sensors indicated in Figure 12 there are usage limitations. For example, adequate lighting is essential for the proper functioning of passive optical systems. Their effectiveness will also depend upon the area of the oil slick, the thickness of the spill layer, and the ice layer underneath the oil slick. For high-performance thermal infrared sensors, it is necessary for there to be a sufficient amount of daylight to heat the oil layer to detect it. It is the stability and homogeneity of the ice layer that affects GPR studies. During seasons with high humidity, when the ice has not yet fully formed, the use of airborne radar stations will be ineffective [67].

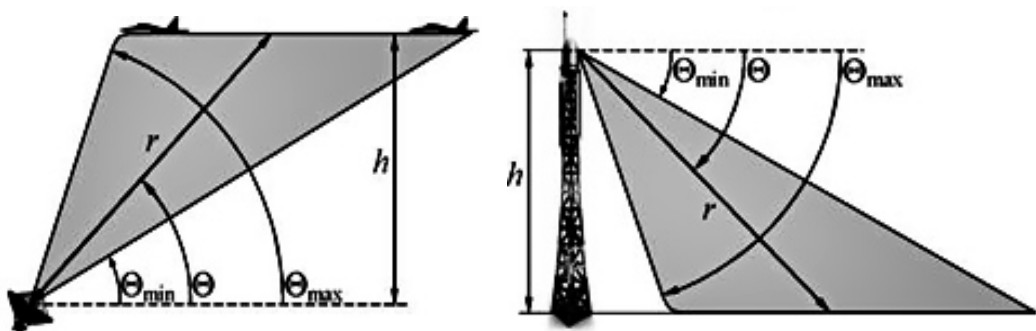

**Figure 12.** Examples of using the cosecance radiation pattern [68].

Above-ice sensors such as radar systems were the only ones capable of detecting oil below or trapped within the ice, but they require further development to lead to an operational system [67]. To achieve greater efficiency when using the radar method for monitoring oil spills, it is necessary to use antenna systems with cosecant radiation patterns (RP). In all practical applications, the need to use a cosecant RP is associated with ensuring equal density of the radio wave energy flow in a wide angular sector of directions from the object (or to the object) of the radio system action [68].

In this case, a special diagram ensures that the display on the monitor screen of targets with different ranges from the radar station will have the same brightness, due to their uniform irradiation. For on-board radio systems, the cosecance RP provides uniform irradiation of the underlying surface, regardless of the angle of approach. This is optimal for basic cellular radio and digital television stations, which require coverage of the service area (Figure 12 right) with a constant level of power at the input of subscribers' receivers [69].

To increase the range of the positioning system, it is necessary to provide a higher signal level from the repeater and reduce the influence of multipath propagation. Due to the formation of a discrete–tunable directional pattern of the antenna array and the use of active antenna elements, the accuracy and noise immunity of the local navigation system can be greatly increased.

The antenna array is composed of rectangular printed antenna elements. The elements are displaced in a plane that is perpendicular to the plane passing through the excitation point of the antenna elements and their centers. The choice of such an arrangement of the elements of the antenna reduces the mutual interference between the elements. When the substrate thickness is 1.5 mm, the relative permittivity is 4.2, and the central operating frequency is 1.5 GHz, the antenna array element dimensions are 42 mm × 42 mm. The feed point is located 13 mm from the edge of the printing element, with an input impedance of 50 ohms. The distance between the edges of the antenna elements is also 42 mm. This is half the wavelength in the dielectric [70].

Figure 13 illustrates the directional diagrams of a three-element antenna array with in-phase and non-in-phase excitation of elements. It is possible to significantly change the position of the minima of the radiation pattern by changing the excitation phases of the side elements relative to the central element. This will decrease the influence of interference on the useful signal [70].

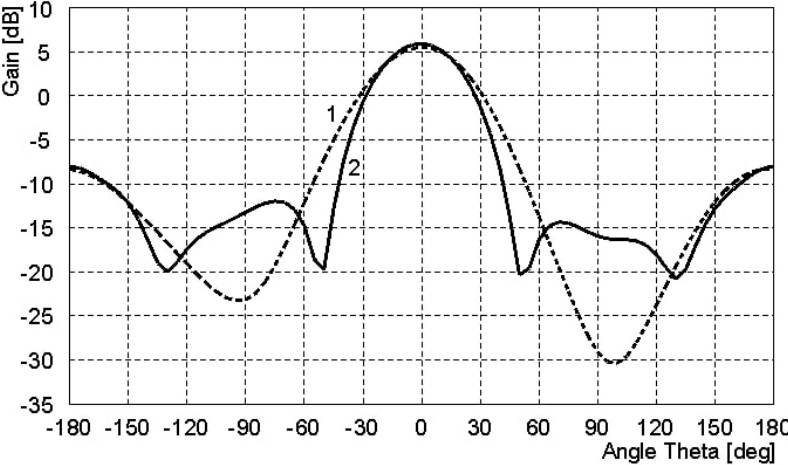

**Figure 13.** Directional diagram of a three-element antenna array (curve 1—in−phase excitation, 2—non−in−phase excitation) [70].

Meanwhile, each element of the antenna array is combined with a single-port mi-crowave amplifier, which makes the array element an active transceiver antenna. Figure 14 shows the results of modeling the transmission coefficient of the microwave signal in the

communication channel with and without an amplifier. With an amplifier, the transmission coefficient was significantly higher [70].

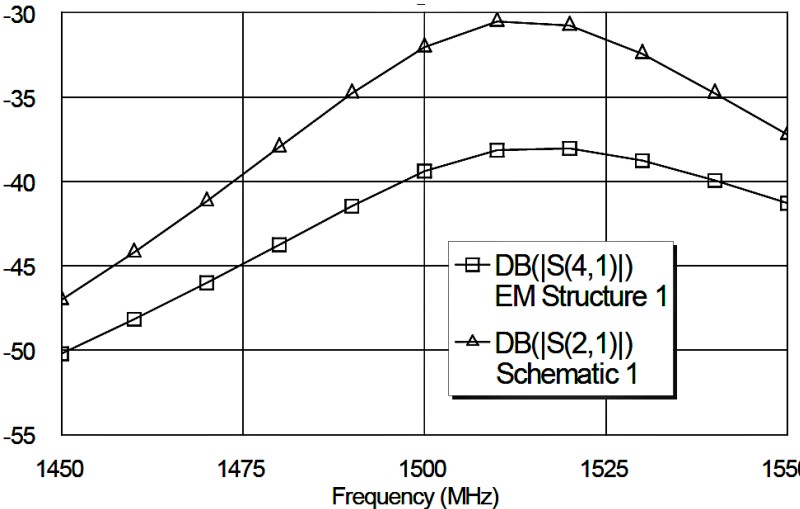

**Figure 14.** Simulation results of microwave signal transmission by passive and active antenna element [70].

The amplifier designed according to this scheme allows one to increase the signal level by 8 dB without being sensitive to the matching of the input resistance.

Since the use of a cosecant radiation pattern and an active antenna array increases the effectiveness of the radar monitoring method as a whole, the method will prove to be far more effective than previous methods. Further articles are planned to explore these advantages in greater detail [71].

## 6. Conclusions

The article described the conditions and features of oil spills in the Arctic region as well as their impact on the environment. Further, there was a discussion of the different types of instruments and methods that could be used for remote monitoring of these possible oil pollutants. As a result, methods using radar, ultraviolet, and infrared were considered to be the most effective for water surface monitoring under such conditions as strong wind, low temperature, and a layer of ice covering the water.

Some sensors may complement each other in terms of oil thickness resolution vs. area coverage or swath width. In the future, operational systems will likely use suites of different sensors operating from various platforms under, on, and above the ice surface to detect oil in a range of ice environments.

The authors intend to undertake further research into the application of the radar monitoring method with the use of a cosecance directional diagram and an active antenna array. Furthermore, it is planned to conduct a series of field experiments to test the theoretical conclusions reached above.

**Author Contributions:** Conceptualization, A.A.K. and V.A.S.; methodology, A.A.K.; validation, A.A.K., K.D.P. and V.A.S.; formal analysis, K.D.P.; investigation, K.D.P.; resources, A.A.K.; data curation, V.A.S.; writing—original draft preparation, K.D.P. and A.A.K.; writing—review and editing, V.A.S.; visualization, K.D.P.; supervision, A.A.K.; project administration, A.A.K.; funding acquisition, A.A.K. All authors have read and agreed to the published version of the manuscript.

**Funding:** This research received no external funding.

**Institutional Review Board Statement:** Not applicable.

**Informed Consent Statement:** Not applicable.

**Data Availability Statement:** All relevant data are included in the manuscript.

**Conflicts of Interest:** The authors declare no conflict of interest.

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
