# Peer review of "Features of Oil Spills Monitoring on the Water Surface by the Russian Federation in the Arctic Region"

_jmse, doi:10.3390/jmse11010111_

Round 1
Reviewer 1 Report
Although this article addresses several topics related to oil spill monitoring in the Arctic, a subject of considerable current interest, the coverage of the individual topics is for the most part superficial and simplistic. The introduction includes two tables of major oil spills worldwide, most of which occurred outside the Arctic, and are partially duplicative. Some of the analytical methods presented as monitoring are so routine (e.g. gravimetric, infrared spectroscopy, gas chromatography) as to be hardly worth the exposition given them, in contrast with the more relevant and interesting remote sensing methods that immediately follow. Some of the results presented in section 5 are intriguing, but lack supporting evidence or references. Critical references are also absent elsewhere in the manuscript, such as for the alleged 700,000 bbl spill volume for the Exxon Valdez spill, the 70% of Arctic oil resources allegedly located offshore, and others.
More generally, little of the material or results presented is new, and most can be found in the recent “Oil in the Sea IV” book published by the US National Academy of Science, Engineering and Medicine. I recommend the authors peruse this book, and if they still feel they have anything new to add with regard to oil spill monitoring in the Arctic, they prepare a new article that focuses narrowly on the new material.
Author Response
Dear Sir or Madam,
Thank you so much for your review of my research. Answers on your remarks are below. New version of the article is in the attachment.
- The introduction was expanded, additional references were added;
- Yes, in the beginning of the research there are mostly examples of worldwide oil spills due to this part of article is talking about the effect of oil spills in the World and in the Arctic region separately. There were added more examples of oil spills in the Arctic region.
- We added new material about research of the radar method and its advantages due to the cosecance radiation pattern and active antenna array in the section №6;
- Moderate English changes were made.
Could you clarify that exactly means this point “Is the research design appropriate”?
Looking forward to hearing from you!

Reviewer 2 Report
1. This paper is more like a review.
2. Remote sensing methods and even remote sensing methods of oil in ice are not fully introduced, such as visible and hyperspectral remote sensing, Lidar, and so on are rarely or not mentioned.
3. FIG. 1 shows little data for the last 20 years, with no examples of polar or ice regions.
4. In FIG. 11 and FIG. 12, low resolution and local blur exist.
5. Lack of data to qualitatively measure the strengths and weaknesses of different approaches.
Author Response
Dear Sir or Madam,
Thank you so much for your review of my research. Answers on your remarks are below. New version of the article is in the attachment.
- Information about remote sensing methods and even remote sensing methods of oil in ice was added in sections 5 and 6;
- There were added more examples of oil spills in the Arctic region;
- The quality of figures was increased;
- All pros and cons of methods are showed in the section 4. We added some new information about advantages of radar method in the section 6;
- Moderate English changes were made.
Looking forward to hearing from you!

Reviewer 3 Report
Dear Authors,
The topic of the article is interesting and intelligible. The manuscript has certainly potential to improve. To help improve the quality of this manuscript, I have added more comments bellow:
- The manuscript should give overview of contemporary available literature and scientific papers on the subject
- Personal form in writing („we analyzed“, „we define“ etc.) should be avoided. The manuscript should be written in inpersonal form and objective.
- All abbreviations should be explained
- The last two sentences prior to the subheading 2.1 should be deleted since they are being repeated as subheading 2.1 and the last sentence in the subheading.
- All figures and tables must be referenced
- Subheading 2.3- the exact date and location of the Klova River accident as well as the mentioned „mixture“ must be specified.
- Superscripts must be used
- Tables 3 & 4, except lacking the reference, are unclear.
- The Conclusion is lacking the final remarks and recommendations for future research.
Author Response
Dear Sir or Madam,
Thank you so much for your review of my research. Answers on your remarks are below. New version of the article is in the attachment.
- The list of references was expanded by contemporary available literature and scientific papers on the subject;
- Personal forms were excluded;
- All abbreviations were explained;
- The last two sentences prior to the subheading 2.1 were deleted;
- All figures and tables are referenced by references in the text;
- Subheading 2.3- the exact date and location of the Klova River accident as well as the mentioned „mixture“ was specified;
- Superscripts were used;
- References on tables 3 & 4 are added in the text;
- Final remarks are added in the conclusion. Some recommendations for future research are added in the section 6;
- Moderate English changes were made.
Could you clarify that exactly means this point “Is the research design appropriate”?
Looking forward to hearing from you!

Round 2
Reviewer 1 Report
While this revision is considerably improved in comparison with the version initially submitted, there are still a few relatively minor issues that need to be addressed.
First and foremost, the title should be revised to read "Features of oil spill monitoring by the Russian Federation in the Arctic region". The main reason this is needed is because the monitoring methods presented in section 4 refer almost entirely to Russian-defined methods, but conversely omit methods that are widely used in the rest of Europe and North America, such as one or two-dimensional gas chromatography with mass spectrometric detection. Also, development of the new radar methods for oil spill detection on or beneath ice in the final section of the paper appears to be a novel (and interesting) Russian development.
Second, a definition of exactly what is meant by "Arctic" should be given at the beginning of section 2.2. The definition implicitly used seems to be the 10 C July mean isotherm, which should be stated explicitly if true.
Elsewhere in section 2.2, references should be supplied for the Selendang Ayu spill, the Baffin Island Oil Spill experiment, the Chedabucto Bay spill, the Bravo platform blowout, and the Antonio Gramsci spill. Also, the 1993 Braer spill should be included with a reference here.
In section 2.4, it isn't clear why the Deepwater Horizon is included, as it's not anywhere near the Arctic by any definition. Also, the sentence that refers to British Petroleum's "careless attitude towards safety issues" is a value judgement that is inappropriate for a scientific manuscript, and in addition may even be libelous.
In the second paragraph of section 3.1, include at least Conoco-Phillips as a major operator in northern Alaska (there are several others as well that operate there). In the fourth paragraph, a more appropriate reference for [27] is:
Gautier DL, Bird KJ, Charpentier RR, Grantz A, Houseknecht DW, Klett TR, Moore TE, Pitman JK, Schenk CJ, Schuenemeyer JH, Sørensen K. Assessment of undiscovered oil and gas in the Arctic. Science. 2009 May 29;324(5931):1175-9.
At the end of sec. 3.1, Figure 1 appears to be duplicated. Also, the figure legends for this and all the subsequent figures are inadequate. Figure legends should include sufficient detail to allow readers to interpret the figure without reference to the manuscript text.
The introductory text that immediately follows the title of sec. 4 refers only to chemical analysis methods, but sec. 4 from subsection 4.5 onwards presents remote sensing methods. Either introduce the remote sensing methods in the introductory material immediately following the sec. 4 title, or create a new section (section 5) the presents the remote sensing methods.
In the second bullet on p. 13 beneath Figure 6, it is stated that "...frequency of monitoring depends on how far the satellite is from earth." Is this because of the relationship between the periodicity with distance of a satellite orbit? Does it apply to stationary orbits, where the satellite remains in a fixed position with regard to the earth?
The final sentence of the third full paragraph on p. 19 states that "The thickness and volume per square kilometer can be determined from the color of the slick." This is news to me, as I'm under the impression that determination of slick thickness and volume remain an area of controversy and active research. At minimum, references should be supplied to support this claim. Also, I find the final paragraph on this page incomprehensible, in part because the legend for Figure 9 conveys minimal information and is itself confusing, as it seems to depict a layer of oil above a layer of ice, but the figure legend implies the opposite. In any case both the text and the figure legend should explain in much more detail what is going on here. Finally, the last sentence of this paragraph needs a reference.
The paragraph on p. 20 also needs a reference, and the right-hand scale on Fig. 11 needs units and an explanation in the figure legend (the rest of the figure needs explanation in the legend as well).
The last three figures of the manuscript, starting on p. 22, are all labeled as "Figure 13".
Additional figures that depict the antenna and other structural elements in the text presenting the details of the active antenna array for the cosecant radar detection system should be supplied to furnish visual context for the discussion of the system details presented. This portion of the manuscript seems particularly novel and interesting, but very hard to follow.
Author Response
Dear Sir or Madam,
Thank you so much for your review of my research. Answers on your remarks are below. New version of the article is in the attachment.
We have added some information about two-dimensional gas chromatography with mass spectrometric detection in the chapter 4.1. Moreover, over methods were described taking into account the experience of international scientists and the information from their papers. Therefore, we offer not to change the title of this paper.
We have added the definition of “Arctic”.
We have added all references, which you have mentioned in your review.
The part about British Petroleum's "careless attitude towards safety issues was deleted. The deepwater horizon is included as a vivid example of the possibility of an accident at such a high-tech facility, and what consequences it can lead to. Examples about Arctic accidents are shown in the chapter 2.2 The Arctic incidents.
New companies were added. In the fourth paragraph, a more appropriate reference for [27] was adeed.
Legends on figures were updated. remote sensing methods
The introductory text that immediately follows the title of sec. 4 was added by the information about remote sensing methods.
Yes, the statement "...frequency of monitoring depends on how far the satellite is from earth." Is about the relationship between the periodicity with distance of a satellite orbit. We have not applied the geostationary orbit as the satellite on this orbit experiences braking on the discharged interplanetary medium, experiences the gravitational influence of other celestial bodies (the Moon to a greater extent). It is too complicated for such cases.
About the states that "The thickness and volume per square kilometer can be determined from the color of the slick." We have added the improvement about microwave radiometer and the reference on the paper about this method.
There was added the reference to the paragraph on the page 20, legend to the figure 11 was added.
We have added some legend and description to the figure 9 to make it more clear.
The cosecant radiation pattern is our new topic for the next research, so in this paper we reflected only the beginning of our research on this topic. More information and maybe patent on such construction will be in the next paper.
Looking forward to hearing from you!

Reviewer 3 Report
Dear authors,
the improvements have been made, especially in English, but there are still some corrections needed:
- All figures and tables must be referenced- the reference number should be put at figures and tables and not at sentences referring to figures and tables. Figure 1, 2, 3, 4, 5, 6, 7, 8, 9, 10, 11, 13, and 14 and tables 1, 2, 3, and 4 lack reference. Figures 1, 9, 11 and 13 are of inadequate quality (they are blurry). Figure 5 lacks the triangle symbol for UAV in figure's title. Figures 7 and 8 contain two same images. Figure 12 is actually a table-it should be Table 5, it is blurry and it lacks reference. There are three figures 13 even though in the text the second and the third Figure 13 are referred as Figure 14 and Figure 15.
- The added part in the subheading 2.2 must be referenced
- The location (country) of the Klova River accident should be specified. Also, the terminology must be uniform- Klova is once written as Klova and the next time as Clova.
- Pp 10- UAV is first mentioned as abbreviation and later written in full
Author Response
Dear Sir or Madam,
Thank you so much for your review of my research. Answers on your remarks are below. New version of the article is in the attachment.
References for pictures and tables were added.
We have improved the quality of figures.
The misprints in numbers of figures were corrected.
The figure 12 was changed into the table 5.
The reference to the topic 2.2. was added.
The country of Kolva river was mentioned in the text.
The using of UAV abbreviation was checked, misprints were corrected.
Looking forward to hearing from you!
